# Cooperative neural networks (CoNN): Exploiting prior independence structure for improved classification

**Harsh Shrivastava** [*]
Georgia Tech
hshrivastava3@gatech.edu

**Eugene Bart** [†]
PARC
bart@parc.com

**Bob Price** [†]
PARC
bprice@parc.com

**Hanjun Dai** [*]
Georgia Tech
hanjundai@gatech.edu

**Bo Dai** [*]
Georgia Tech
bodai@gatech.edu

**Srinivas Aluru** [*]
Georgia Tech
aluru@cc.gatech.edu

## Abstract

We propose a new approach, called cooperative neural networks (CoNN), which uses a set of cooperatively trained neural networks to capture latent representations that exploit prior given independence structure. The model is more flexible than traditional graphical models based on exponential family distributions, but incorporates more domain specific prior structure than traditional deep networks or variational autoencoders. The framework is very general and can be used to exploit the independence structure of any graphical model. We illustrate the technique by showing that we can transfer the independence structure of the popular Latent Dirichlet Allocation (LDA) model to a cooperative neural network, CoNN-sLDA. Empirical evaluation of CoNN-sLDA on supervised text classification tasks demonstrates that the theoretical advantages of prior independence structure can be realized in practice - we demonstrate a 23% reduction in error on the challenging MultiSent data set compared to state-of-the-art.

## 1 Introduction

Neural networks offer a low-bias solution for learning complex concepts such as the linguistic knowledge required to separate documents into thematically related classes. However, neural networks typically start with a fairly generic structure, with each level comprising a number of functionally equivalent neurons connected to other layers by identical, repetitive connections. Any structure present in the problem domain must be learned from training examples and encoded as weights. In practice, some domain structure is often known ahead of time; in such cases, it is desirable to pre-design a network with this domain structure in mind. In this paper, we present an approach that allows incorporating certain kinds of independence structure into a new kind of neural learning machine.

The proposed approach is called "Cooperative Neural Networks" (CoNN). This approach works by constructing a set of neural networks, each trained to output an embedding of a probability distribution. The networks are iteratively updated so that each embedding is consistent with the embeddings of the other networks and with the training data. Like probabilistic graphical models, the representation is factored into components that are independent. Unlike probabilistic graphical

---

[*]Dept. of Comp. Sci. & Eng. Georgia Institute of Technology Atlanta, GA 30332
[†]3333 Coyote Hill Rd, Palo Alto, CA,

models, which are limited to tractable conditional probability distributions (e.g., exponential family), CoNNs can exploit powerful generic distributions represented by non-linear neural networks. The resulting approach allows us to create models that can exploit both known independence structure as well as the expressive powers of neural networks to improve accuracy over competing approaches.

We illustrate the general approach of cooperative neural networks by showing how one can transfer the independence structure from the popular Latent Dirichlet Allocation (LDA) model [2] to a set of cooperative neural networks. We call the resultant model CoNN-sLDA. Cooperative neural networks are different from feed forward networks as they use back-propagation to enforce consistency across variables within the latent representation. CoNN-sLDA improves over LDA as it admits more complex distributions for document topics and better generalization over word distributions. CoNN-sLDA is also better than a generic neural network classifier as the factored representation forces a consistent latent feature representation that has a natural relationship between topics, words and documents. We demonstrate empirically that the theoretical advantages of cooperative neural networks are realized in practice by showing that our CoNN-sLDA model beats both probabilistic and neural network-based state-of-the-art alternatives. We emphasize that although our example is based on LDA, the CoNN approach is general and can be used with other graphical models, as well as other sources of independence structure (for example, physics- or biology-based constraints).

## 2    Related Work

Text classification has a long history beginning with the use of support vector machines on text features [11]. More sophisticated approaches integrated unsupervised feature generation and classification in models such as sLDA [17, 6] and discriminative LDA (discLDA)  [13] and a maximum margin based combination [33].

One limitation of LDA-based models is that they pick topic distributions from a Dirichlet distribution and cannot represent the joint probability of topics in a document ( i.e., hollywood celebrities, politics and business are all popular categories, but politics and business appear together more often than their independent probabilities would predict). Models such as pachinko allocation [15] attempt to address this with complex tree structured priors. Another limitation of LDA stems from the fact that word topics and words themselves are selected from categorical distributions. These admit arbitrary empirical distributions over tokens, but don't generalize what they learn. Learning about the topic for the token "happy" tells us nothing about the token "joyful".

There have been many generative deep learning models such as Deep Boltzmann Machines [27], NADE [14, 32], variational auto-encoders (VAEs) [31] and variations [18], GANs[9] and other deep generative networks [28, 1, 22, 20] which can capture complex joint distributions of words in documents and surpass the performance of LDA. These techniques have proven to be good generative models. However, as purely generative models, they need a separate classifier to assign documents to classes. As a result, they are not trained end-to-end for the actual discriminative task that needs to be performed.  Therefore, the resulting representation that is learned does not incorporate any problem-specific structure, leading to limited classification performance. Supervised convolutional networks have been applied to text classification [12] but are limited to small fixed inputs and still require significant data to get high accuracy.  Recurrent networks have also been used to handle open ended text [8].  A supervised approach for LDA with DNN was developed by [4, 5] using end-to-end learning for LDA by using Mirror-Descent back propagation over a deep architecture called BP-sLDA. To achieve better classification, they have to increase the number of layers of their model, which results in higher model complexity, thereby limiting the capability of their model to scale. In summary, there are still significant challenges to creating expressive, but efficiently trainable and computationally tractable models.

In the face of limited data, regularization techniques are an important way of trying to reduce overfitting in neural approaches. The use of pretrained layers for networks is a key regularization strategy; however, training industrial applications with domain specific language and tasks remains challenging. For instance, classification of field problem reports must handle content with arcane technical jargon, abbreviations and phrasing and be able to output task specific categories.

Techniques such as L2 normalization of weights and random drop-out [26] of neurons during training are now widely used but provide little problem specific advantage. Bayesian neural networks with distributions have been proposed, but independent distributions over weights result in network

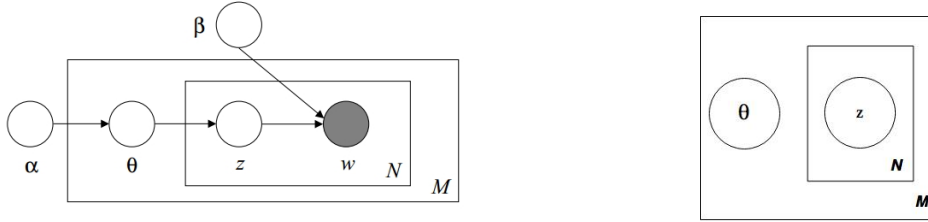

(a) LDA summarizes the content of each document $m$ in $M$ as a topic distribution $\theta_m$. Each word $w_{m,n}$ in $N_m$ has topic $z_{m,n}$ drawn from $\theta_m$.

(b) Variational LDA approximates the posterior topic distribution $\theta_m$ and word topic $z_{m,n}$ with independent distributions.

Figure 1: Plate models representing the original LDA and its approximation.

weight means where the variance must be controlled fairly closely so that relative relationship of weights produces the desired computation. Variational auto-encoders explicitly enable probability distributions and can therefore be integrated over, but are still largely undifferentiated structure of identical units. They don't provide a lot of prior structure to assist with limited data.

Recently there has been work incorporating other kinds of domain inspired structure into networks such Spatial transformer networks [10], capsule networks [23] and natural image priors [21].

## 3 Deriving Cooperative Neural Networks

Application of our approach proceeds in several distinct steps. First, we define the independence structure for the problem. In our supervised text classification example, we incorporate structure from latent dirichlet allocation (LDA) by choosing to factor the distribution over document texts into document topic probabilities and word topic probabilities. This structure naturally enforces the idea that there are topics that are common across all documents and that documents express a mixture of these topics independently through word choices. Second, a set of inference equations is derived from the independence structure. Next, the probability distributions involved in the variational approximation, as well as the inference equations, are mapped into a Hilbert space to reduce limitations on their functional form. Finally, these mapped Hilbert-space equations are approximated by a set of neural networks (one for each constraint), and inference in the Hilbert space is performed by iterating these networks. We call the combination of Cooperative Neural Networks and LDA as Cooperative Neural Network supervised Latent Dirichlet Allocation, or 'CoNN-sLDA'. These steps are elaborated in the following sections.

### 3.1 LDA model

Here, we use the same notation and the same plate diagram (Figure 1a) as in the original LDA description [2]. Let $K$ be the number of topics, $N$ be the number of words in a document, $V$ be the vocabulary size over the whole corpus, and $M$ be the number of documents in the corpus. Given the prior over topics $\alpha$ and topic word distributions $\beta$, the joint distribution over the latent topic structure $\theta$, word topic assignments $\mathbf{z}$, and observed words in documents $\mathbf{w}$ is given by:

$$p(\theta, \mathbf{z}, \mathbf{w}|\alpha, \beta) = p(\theta|\alpha)\prod_{i=1}^{N} p(z_i|\theta)p(w_i|z_i, \beta) \tag{1}$$

### 3.2 Variational approximation to LDA

Inference in LDA requires estimating the distribution over $\theta$ and $\mathbf{z}$. Using the Bayes rule, this posterior can be written as follows:

$$p(\theta, \mathbf{z}|\mathbf{w}, \alpha, \beta) = \frac{p(\theta, \mathbf{z}, \mathbf{w}|\alpha, \beta)}{p(\mathbf{w}|\alpha, \beta)} \tag{2}$$

Unfortunately, directly marginalizing out $\theta$ in the original model is intractable. Variational approximation of $p(\theta, \mathbf{z})$ is a common work-around. To perform variational approximation, we approximate

this LDA posterior with the Probabilistic Graphical Model (PGM) shown in Figure 1b. The joint distribution for the approximate PGM is given by:

$$q(\theta, \mathbf{z}) = q(\theta) \prod_{i=1}^{N} q_i(z_i) \tag{3}$$

We want to tune the approximate distribution to resemble the true posterior as much as possible. To this end, we minimize the KL divergence between the two distributions. Alternatively, this can be seen as minimizing the variational free energy of the Mean-Field inference algorithm [30]:

$$\min_{\{q\}} \{ D_{KL}( \, q(\theta, \mathbf{z}) \, || \, p(\theta, \mathbf{z} | \mathbf{w}, \alpha, \beta) \, ) \} \tag{4}$$

To solve this minimization problem, we derive a set of fixed-point equations in Appendix(A). These fixed-point equations can be expressed as

$$\log q(\theta) = \log p(\theta | \alpha) + \sum_{i=1}^{N} \int_{z_i} q_i(z_i) \log p(z_i | \theta) \, dz_i - 1 \tag{5}$$

$$\log q_i(z_i) = \log p(w_i | z_i, \beta) + \int_{\theta} q(\theta) \log p(z_i | \theta) d\theta - 1 \tag{6}$$

This set of equations is difficult to solve analytically. In addition, even if it was possible to solve them analytically, they are still subject to the limitations of the original graphical models, such as the need to use exponential family distributions and conjugate priors for tractability.

Therefore, the next step in the proposed method is to map the probability distributions and the corresponding fixed-point equations into a Hilbert space, where some of these limitations can be relaxed. Section 3.3 gives a general overview of Hilbert space embeddings, and section 3.4 derives the corresponding equations for our model.

### 3.3 Hilbert Space Embeddings of Distributions

We follow the notations and procedure defined in [7] for parameterizing Hilbert spaces. By definition, the Hilbert Space embeddings of probability distributions are mappings of these distributions into potentially *infinite* -dimensional feature spaces. [24]. For any given distribution $p(X)$ and a feature map $\phi(x)$, the embedding $\mu_X : \mathcal{P} \to \mathcal{F}$ is defined as:

$$\mu_X \; := \; E_X[\phi(X)] \; = \; \int_{\mathcal{X}} \phi(x) p(x) dx \tag{7}$$

For some choice of feature map $\phi$, the above embedding of distributions becomes injective [25]. Therefore, any two distinct distributions $p(X)$ and $q(X)$ are mapped to two distinct points in the feature space. We can treat the injective embedding $\mu_X$ as a sufficient statistic of the corresponding probability density. In other words, $\mu_X$ preserves all the information of $p(X)$. Using $\mu_X$, we can uniquely recover $p(X)$ and any mathematical operation on $p(X)$ will have an equivalent operation on $\mu_X$. These properties lead to the following equivalence relations. We can compute a functional $f : \mathcal{P} \to \mathbb{R}$ of the density $p(X)$ using only its embedding,

$$f(p(x)) = \tilde{f}(\mu_X) \tag{8}$$

by defining $\tilde{f} : \mathcal{F} \to \mathbb{R}$ as the operation on $\mu_X$ equivalent to $f$. Similarly, we can generalize this property to operators. An operator $\mathcal{T} : \mathcal{P} \to \mathbb{R}^{\mathrm{d}}$ applied to a density can also be equivalently carried out using its embedding,

$$\mathcal{T} \circ p(x) = \tilde{\mathcal{T}} \circ \mu_X \tag{9}$$

where $\tilde{\mathcal{T}} : \mathcal{F} \to \mathbb{R}^{\mathrm{d}}$ is again the corresponding equivalent operator applied to the embedding. In our derivations, we assume that there exists a feature space where the embeddings are injective and apply the above equivalence relations in subsequent sections.

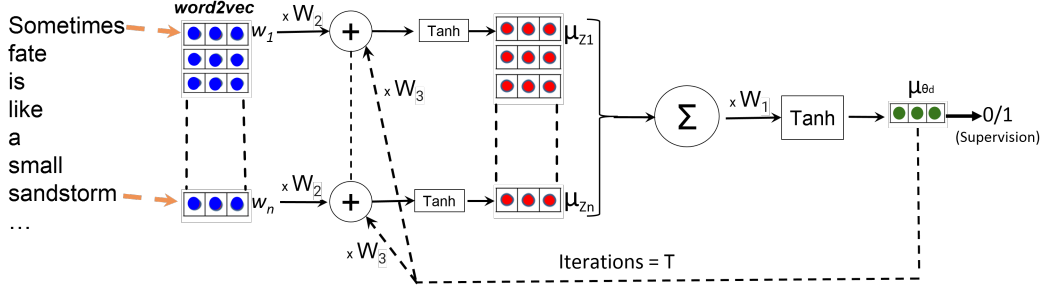

Figure 2: Visualization of the CoNN-sLDA architecture for a single document. For the $i$'th word, the latent topic variable is $z_i$. The embedding for the distribution $p(z_i)$ is $\mu_{z_i}$; these embeddings are shown as three-dimensional vectors for illustration. They are accumulated and passed through a non-linearity to obtain $\mu_\theta$, which is the embedding of $p(\theta)$, the distribution over the topics for the document. Thus, the embedding $\mu_\theta$ is determined (up to the non-linearity) by the average of the embeddings $\mu_{z_i}$, as in the original LDA model. Similarly, there is feedback from $\mu_\theta$ (which happens for $T$ iterations, see Alg1), so that $\mu_\theta$, in turn, influences $\mu_{z_i}$, again, as in the original LDA model.

### 3.4 Hilbert space embedding for LDA

We consider Hilbert space embeddings of $q(\theta)$, $q_i(z_i)$, as well as the equations (5) and (6). By definition given in equation(7),

$$\mu_\theta = \int_\theta \phi(\theta)q(\theta)d\theta \qquad \mu_{z_i} = \int_{z_i} \phi(z_i)q_i(z_i)dz_i \tag{10}$$

The variational update equations in (5) and (6) provide us with the key relationships between latent variables in the model. We can replace the specific distributional forms in these equations with operators that maintain the same relationships among distributions represented in the Hilbert space embeddings.

$$q(\theta) = f_1(\theta, \{q_i(z_i)\}) \quad q_i(z_i) = f_2(z_i, w_i, q(\theta)) \tag{11}$$

Here, $f_1$ and $f_2$ represent the abstract structure of the model implied by (5) and (6) without specific distributional forms. We will provide a specific instantiation of $f_1$ and $f_2$ shortly. Following the same argument as in equation (8), we can write equation (11) as $q(\theta) = \tilde{f}_1(\theta, \{\mu_{z_i}\})$. Similarly, $q_i(z_i) = \tilde{f}_2(z_i, w_i, \mu_\theta)$. Iterating through all values of $\theta, z_i$ and using the operator view given in equation (9) as reference, we get the following equivalent fixed-point equations in the Hilbert Space:

$$\mu_\theta = \mathcal{T}_1 \circ \{\mu_{z_i}\} \qquad \mu_{z_i} = \mathcal{T}_2 \circ [w_i, \mu_\theta] \tag{12}$$

### 3.5 Parameterization of Hilbert space embedding using Deep Neural Networks

The operators $\mathcal{T}_1$ and $\mathcal{T}_2$ have complex non-linear dependencies on the unknown true probability distributions and the feature map $\phi$. Thus, we need to model these operators in such a way that we can utilize the available data to learn the underlying non-linear functions. We will use deep neural networks which are known for their ability to model non-linear functions.

We start by parameterizing the embeddings. We assume that any point in the Hilbert space is a vector $\mu_i \in \mathbb{R}^D$. Next, as the operators are non-linear function maps, we replace them by deep neural networks. In its simplest form, we only use a single fully connected layer with 'tanh' activations yielding the following fixed point update equations,

$$\mu_\theta = \tanh(\ W_1 \cdot \sum_{i=1}^{N} \{\mu_{z_i}\}\ ) \tag{13}$$

$$\mu_{z_i} = \tanh(\ W_2 \cdot word2vec(w_i) + W_3.\mu_\theta\ ) \tag{14}$$

The original work on Hilbert space embeddings required the embeddings to be injective. We observe that we do not need the embedding to be injective on the domain of all distributions. Instead, we only

need it to be injective on the sub-domain of distributions used in the training corpus. The supervised training process on the training set will have to find embeddings that allow the model to distinguish documents that occur in the corpus automatically causing the learned embeddings to be injective for the training domain.

We keep the dimension of the $word2vec$ [19] embedding identical to the Hilbert space embedding, i.e. $w_i \in \mathrm{I\!R}^D$. Note, that the above parameterization is one example. Multiple fully connected layers can be used to achieve denser models.

Assume the parameters $word2vec$, $W_1$, $W_2$ and $W_3$ are known. We calculate the set of embeddings for a given text corpus by iterating equations(13, 14). Algorithm 1 summarizes this procedure. We normalize the embeddings after every iteration to avoid overflow. This is the heart of the Cooperative Neural Network paradigm in which a set of neural networks co-constrain each other to produce an embedding informed by prior structure. In our experience, we found that 'tanh' works better than '$\sigma$' as a choice for non-linearity. Using rectified linear 'ReLU' units will not work as they zero out negative values of the embeddings. We apply dropout [26] to $\mu_{z_i}$'s, $\mu_\theta$ and $word2vec$ for regularization. For every document, the algorithm returns the associated $\mu_\theta$ embedding, representing the document in the Hilbert space.

---

**Algorithm 1** Getting Hilbert Space Embeddings

---

**Input:** Parameters $\{W_1, W_2, W_3\}$
Initialize $\{\mu_\theta^{(0)}, \mu_{z_i}^{(0)}\} = \mathbf{0} \in \mathrm{I\!R}^D$.
**for** $t = 1$ **to** T iterations **do**
    **for** $i = 1$ **to** $N$ words **do**
        $\mu_{z_i}^{(t)} = \tanh(W_2.word2vec(w_i) + W_3.\mu_\theta^{(t)})$
        Normalize $\mu_{z_i}^{(t)}$
    **end for**
    $\mu_\theta^{(t)} = \tanh(W_1.\sum_{i=1}^{N}\{\mu_{z_i}^{(t-1)}\})$
    Normalize $\mu_\theta^{(t)}$
**end for**
return $\{\mu_\theta^{(T)}\}$ : Document embeddings

---

**Algorithm 2** Training using Hilbert Space Embeddings

---

**Input:** Document Corpus $\mathcal{D}$, with each doc '$d$' has set of words $[w_{d,i}] \in N_d$.
*Initialize* $\mathbf{P^{(0)}} = \{\mathbf{W^{(0)}}, \mathbf{u^{(0)}}, \mathbf{word2vec^{(0)}}\}$ with random values. Let 'learning rate = $r$'.
**for** $t = 1$ **to** $\mathcal{T}$ **do**
    Sample docs from $\mathcal{D}$ as $\{D_s, y_s\}$
    Using Alg(1) get Hilbert embeddings $\{\mu_{\theta_d}^s\}$ for '$D_s$'
    $y_{pred} = \mathcal{H}\left(\mu_{\theta_d}^s; \mathbf{P^{(t-1)}}\right)$
    Update: $\mathbf{P^{(t)}} = \mathbf{P^{(t-1)}} - r. \bigtriangledown_{\mathbf{P^{(t-1)}}} L(y_{pred}, y_s)$
**end for**
return $\{\mathbf{P^{\mathcal{T}}}\}$

---

In practice, the parameters $word2vec$, $W_1$, $W_2$ and $W_3$ are not known and need to be learned from training data. This requires formulating an objective function, and then optimizing that objective function. An additional advantage of the proposed method is that it allows using a wide variety of objective functions. In our case, we trained the model using a discriminative/supervised criterion that relies on the labels associated with each document, and we used binary cross-entropy loss or cross-entropy loss for multiclass classification.

Algorithm 2 summarizes the training procedure. It uses Algorithm 1 as a subroutine. The $\mathcal{H}$ function is chosen to be a single fully connected layer in our implementation, which transforms the input embedding to a vector corresponding to number of classes. We sample (without replacement) a batch of documents $D_s$ from the corpus, compute their embeddings and update the parameters. The loss function takes in the $\mu_\theta$ embeddings and the corresponding document labels. The resulting model, called 'CoNN-sLDA' is schematically illustrated in Figure 2.

The CoNN-sLDA model retains the overall structure of the LDA model by separating the problem into document topic distributions and word topic distributions within each document. As with traditional LDA, one can visualize a document corpus by projecting topic vectors associated with documents into a 2D plane (e.g., using MDS, tSNE). An advantage of CoNN-sLDA over typical neural network approaches is that typical DNNs produce only a single embedding, whereas CoNN-sLDA elegantly factors the local and global information into separate parts of the model. An advantage of CoNN-sLDA over traditional probabilistic graphical models is that we can use low-bias, highly expressive distributions implied by the neural network implementations of update operators.

# 4 Experiments

## 4.1 Description of Datasets

We evaluated our model 'CoNN-sLDA' on two real-world datasets. The first dataset is a multi-domain sentiment dataset (MultiSent) [3], consisting of 342,104 Amazon product reviews on 25 different types of products (apparels, books, DVDs, kitchen appliances, $\cdots$). For each review, we go through the ratings given by the customer (between 1 to 5 stars) and label a it as positive, if the rating is higher than 3 stars and negative otherwise. We pose this as a binary classification problem. The average length of reviews is roughly 210 words after preprocessing the data. The ratio of positive to negative reviews is $\sim 8:1$. We use 5-fold cross validation and report the average area under the ROC curve (AUC), in %.

The second dataset is the 20 Newsgroup dataset[3]. It has around 19,000 news articles, divided roughly equally into 20 different categories. We pose this as a multiclass classification problem and report accuracy over 20 classes. The dataset is divided into training set (11,314 articles) and test set (7,531 articles), approximately maintaining the relative ratio of articles of different categories. The average length of documents after preprocessing is $\sim 160$ words. This task becomes challenging as there are some categories that are highly similar, making their separation difficult. For example, the categories "PC hardware" and "Mac hardware" have quite a lot in common.

We apply standard text preprocessing steps to both datasets. We convert everything to lower case characters and remove the standard stopwords defined in the 'Natural Language Toolkit' library. We remove punctuations, followed by lemmatization and stemming to further clean the data. However, for other classifiers, we use the preprocessing techniques recommended by the respective authors.

## 4.2 Baselines for comparison

We compare 'CoNN-sLDA' with existing state-of-the-art algorithms for document classification. We compare against VI-sLDA, [6, 17], which includes the label of the document in the graphical model formulation and then maximizes the variational lower bound. Different from VI-sLDA, the supervised topic model using DiscLDA [13] reduces the dimensionality of topic vectors $\theta$ for classification by introducing a class-dependent linear transformation.

Boltzmann Machines are traditionally used to model distributions and with the recent development of deep learning techniques, these approaches have gained momentum. We compare with one such Deep Boltzmann Machine developed for modeling documents called Over-Replicated Softmax (OverRep-S) [27]. Another popular approach is by [4], called BP-sLDA, which does end-to-end learning of LDA by mirror-descent back propagation over a deep architecture. We also compare with a recent deep learning model developed by [5] called DUI-sLDA.

## 4.3 Classification Results

Table(1) shows the accuracy results on newsgroup dataset together with standard error on the mean (SEM) over 5 folds. For each of 5 folds, we split training data into train and validation and optimize all parameters. We then evaluate against a fixed common test set. As the number of classes is 20, we found that using higher Hilbert space dimensions work better (See entries for Dim=40 and Dim=80 in table). A dropout of $\sim 0.8$ was applied to *word2vec* embeddings. The batch size was fixed at 100 and we trained for around 400 batches. The performance of CoNN-sLDA is better than BP-sLDA and at par with 5 layer DUI-sLDA model. The cost sensitive version CoNN-sLDA (Imb), balances out the misclassification cost for different classes in the loss function tends to perform slightly better. The 20 newsgroup dataset is one of the earliest and most studied text corpuses. It is fairly separable, so most modern state-of-the-art methods do well on it, but it is an important benchmark to establish the credibility of an algorithm.

Our CoNN-sLDA model was able to outperform the recently proposed state-of-the-art method, DUI-sLDA, on the large 'MultiSent' dataset (table2) having over 300K documents by a significant AUC margin of 2%. This corresponds to a 23% reduction in error rate. We used a single fully connected layer with $\tanh$ non-linear function for both, $\mu_\theta, \mu_{z_i}$ embeddings. Hilbert space dimension and

| Classifier | Accuracy(%) | Details |
|---|---|---|
| VI-sLDA | 73.8± 0.49 | $K$=50 |
| DiscLDA | 80.2± 0.45 | $K$=50 |
| OverRep-S | 69.5± 0.36 | $K$=512 |
| BP-sLDA | 81.8± 0.36 | $K$=50, $L$=5 |
| DUI-sLDA | 83.5± 0.22 | $K$=50, $L$=5 |
| CoNN-sLDA | 83.4 ± 0.18 | Dim=40 |
| CoNN-sLDA(imb) | **83.7± 0.13** | Dim=80 |

Table 1: '20 Newsgroups' classification accuracy on 19K documents. SEM over 5 fold CV. Dim indicates Hilbert space dimension.

| Classifier | AUC (%) | Details |
|---|---|---|
| VI-sLDA | 76.8± 0.40 | $K$=50 (topics) |
| DiscLDA | 82.1± 0.40 | $K$=50 |
| BP-sLDA | 88.9± 0.36 | $K$=50, $L$=5 |
| DUI-sLDA | 86.0± 0.31 | $K$=50, $L$=1 |
| DUI-sLDA | 91.4± 0.27 | $K$=50, $L$=5 |
| CoNN-sLDA | **93.3± 0.13** | Dim=10 |
| CoNN-sLDA(imb) | **93.4± 0.13** | Dim=20 |

Table 2: 'MultiSent' AUC on 324K documents. SEM over 5 Fold CV. Dim indicates Hilbert space dimension.

*word2vec* dimension are both 10. We use a dropout probability of 0.1, The Algorithm(1) was unrolled for 1 iteration. 'Batch size' was set at 100 and ran for 3000 batches with optimization done using 'Adam' optimizer. We also ran a cost sensitive version of CoNN-sLDA (Imb) model, with a balancing ratio of 1.4 towards the minority class which was incorporated in the loss function. We observe slight improvement in results. CoNN-sLDA consistently outperformed other models over various choices of model parameters, see Appendix(B).

The number of layers required by other deep models like DUI-sLDA, BP-sLDA for good classification is usually quite high and their performance decreases considerably with fewer layers. CoNN-sLDA outperforms them with a single layer neural network.

We have a vectorized and efficient implementation of CoNN-sLDA in PyTorch and Tensorflow. The results shown above are from the PyTorch version. We ran our experiments on NVIDIA Tesla P100 GPUs. The runtime for 1 fold of 'MultiSent' for the settings mentioned above is around *5 minutes*, while a single fold for '20 Newsgroup' dataset runs within *2 minutes*.

In Appendix(B), we report our experiments to optimize the algorithmic and architectural hyperparameters. We use the 'MultiSent' data for our analysis. In general for training, we recommend starting with a small Hilbert space dimension and batch size, then try increasing the number of fully connected layers and finally choose to unroll the model further.

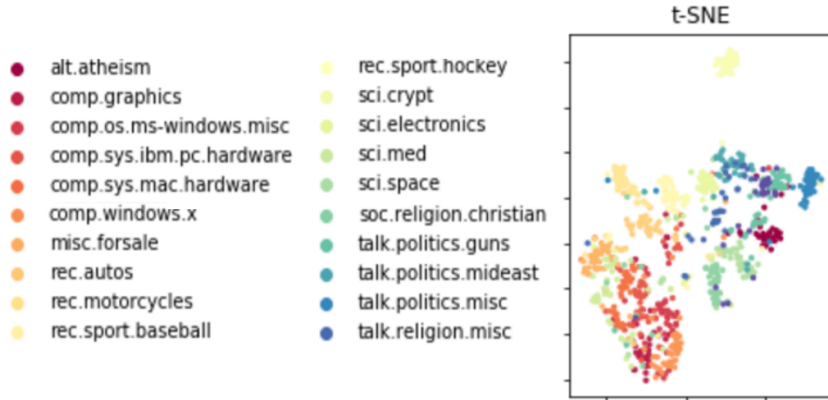

Figure 3: A t-SNE projection of the 40-dimensional embeddings $\mu_\theta$ for test documents in the 20-Newsgroups dataset. The colors represent the category label for each document. The embeddings separate categories very well.

## 5 Discussions & Future extensions

In addition to supervised classification, we can use LDA style models for visualizing and interpreting the cluster structure of the datasets. For example, in CoNN-sLDA model, we can use t-SNE [16] to visualize the documents using their $\mu_\theta$ values. In Figure 3 we see that CoNN-sLDA clearly maps different newsgroups to homogeneous regions of space that help classification accuracy and provide

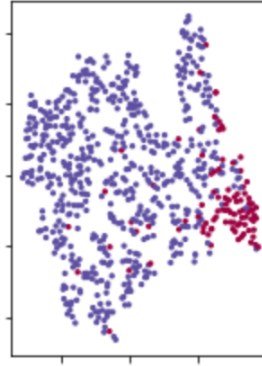

Figure 4: tSNE visualization of a random sample of 10-dimensional $\mu_\theta$ embeddings for Multi-sent documents (Blue positive, red negative). The embeddings project distinct categories to highly coherent regions.

insight into the structure of the domain. Similarly, Figure 4 shows that CoNN-sLDA maps the positive and negative product reviews into different regions facilitating classification and interpretation.

An interesting extension for the CoNN-sLDA model will be to map the Hilbert space topic embedding $\mu_\theta$ back to the original topic space distribution. This would potentially allow us to provide text labels for the discovered clusters providing an intuitive interpretation for the model learned by our technique. Appendix (C) discusses an approach to get most relevant words in a document pertaining to a discriminative task.

In this work, we obtain the fixed point update equations using the mean-field inference technique. In general, we can extend this procedure to other variational inference techniques. For example, we can find embeddings for Algorithm 1 by minimizing the free energies of loopy belief propagation or its variants (e.g., [29]) and use Algorithm 2 to train them end-to-end.

## 6    Conclusion

Cooperative neural networks (CoNN) are a new theoretical approach for implementing learning systems which can exploit both prior insights about the independence structure of the problem domain and the universal approximation capability of deep networks. We make the theory concrete with an example, CoNN-sLDA, which has superior performance to both prior work based on the probabilistic graphical model LDA and generic deep networks. While we demonstrated the method on text classification using the structure of LDA, the approach provides a fully general methodology for computing factored embeddings using a set of highly expressive networks. Cooperative neural networks thus expand the design space of deep learning machines in new and promising ways.

## Acknowledgements

We are thankful to our colleagues Srinivas Eswar, Patrick Flick and Rahul Nihalani for their careful reading of our submission.

## Footnotes

[3] http://qwone.com/ jason/20Newsgroups/

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
