[Supplementary Material]

# Appendix

## A. Derivation of fixed point equations

Inference in LDA requires estimating the distribution over $\theta$ and $\mathbf{z}$. Using the Bayes rule, this posterior can be written as follows:

$$p(\theta, \mathbf{z}|\mathbf{w}, \alpha, \beta) = \frac{p(\theta, z, w|\alpha, \beta)}{p(w|\alpha, \beta)} \tag{15}$$

To perform variational approximation, we approximate this LDA posterior with the PGM as shown in Figure 1b.

The joint distribution for the approximate PGM is given by:

$$q(\theta, z) = q(\theta) \prod_{i=1}^{N} q_i(z_i) \tag{16}$$

We want to tune the approximate distribution to resemble the true posterior as much as possible. To this end, we minimize the KL divergence between the two distributions. Alternatively, this can be seen as minimizing the variational free energy of the Mean-Field inference algorithm [30]:

$$\min_{\{q\}} \{ D_{KL}( \, q(\theta, z) \, || \, p(\theta, z|w, \alpha, \beta) \, ) \} \tag{17}$$

Substituting the expression for KL-divergence, we get

$$\min_{\{q\}} \int_\theta \int \cdots \int_{\{z_i\}} q(\theta, z) \, \log \frac{q(\theta, z)}{p(\theta, z|w, \alpha, \beta)} \, d\theta \, \{dz_i\} \tag{18}$$

Using the Bayes formulation given in equation(2) and observing that $p(w|\alpha, \beta)$ is a constant, we can write

$$\min_{\{q\}} \int_\theta \int \cdots \int_{\{z_i\}} q(\theta, z) \, [ \, \log q(\theta, z) - \log p(\theta, z, w|\alpha, \beta) \, ] \, d\theta \, \{dz_i\} \tag{19}$$

Substituting the probability densities given in equations (1) and (3), the following minimization expression is obtained:

$$\min_{\{q\}} \int_\theta \int_{\{z_i\}} \left\{ q(\theta) \prod_{i=1}^{N} q_i(z_i) \right\} \left\{ \log \left( q(\theta) \prod_{i=1}^{N} q_i(z_i) \right) \right.$$
$$\left. - \log \left( p(\theta|\alpha) \prod_{i=1}^{N} p(z_i|\theta) p(w_i|z_i, \beta) \right) \right\} \, d\theta \, \{dz_i\} \tag{20}$$

Pulling logarithms inwards we can convert products to summations. We then move integrals inward. In some cases, integrals add up to 1 ( $e.g.$, $\int_\theta q(\theta) \, d\theta = 1$). In some cases, inner sums can be pulled outwards. The result consists of simple integrals:

$$\min_{\{q\}} \left\{ \int_\theta q(\theta) \log q(\theta) \, d\theta + \sum_{i=1}^{N} \int_{z_i} q_i(z_i) \log q_i(z_i) \, dz_i - \int_\theta q(\theta) \log p(\theta|\alpha) \, d\theta \right.$$
$$\left. - \sum_{i=1}^{N} \iint_{\theta, z_i} q(\theta) q_i(z_i) \log p(z_i|\theta) \, d\theta \, dz_i - \sum_{i=1}^{N} \int_{z_i} q_i(z_i) \log p(w_i|z_i, \beta) \, dz_i \right\} \tag{21}$$

We denote the expression given in equation(21) by $\min_{\{q\}}(L)$. To minimize the functional equation given by $L$, we take the functional derivatives of $L$ with respect to $q(\theta)$ and $q_i(z_i)$ and equate them to zero.

Solving for $\left( \frac{\delta L}{\delta q(\theta)} = 0 \right)$, we get the first fixed point equation:

$$\log q(\theta) = \log p(\theta|\alpha) + \sum_{i=1}^{N} \int_{z_i} q_i(z_i) \log p(z_i|\theta) \, dz_i - 1 \tag{22}$$

Similarly, solving for $\frac{\delta L}{\delta q_i(z_i)} = 0$, we get the second fixed point equation:

$$\log q_i(z_i) = \log p(w_i|z_i, \beta) + \int_\theta q(\theta) \log p(z_i|\theta) d\theta - 1 \tag{23}$$

Note that this derivation is different from the classical variational approximation derivations, where the EM algorithm is eventually used to iteratively approximate the posterior.

## B. Architecture choices of CoNN-sLDA model

In this section we report on our experiments to optimize the algorithmic and architectural hyperparameters. We use the 'MultiSent' dataset for our analysis.

### B.1 Varying Hilbert Space Embeddings dimension

Figure 5: Varying Hilbert space embeddings dimension along the x-axis and the AUC values on y-axis. We also compare with the cost sensitive learning version, denoted by AUC (IMB) for every dimension choice. The depth of neural networks for both the embeddings $\mu_\theta$ and $\mu_{z_i}$ is a single fully connected layer.

The dimensionality of the Hilbert space trades off the expressive power against computation and storage requirements of the model. In Figure 5, we show varying Hilbert space dimensions on the x-axis and compare their AUCs. We observe a decline in AUC after Hilbert dimension of 20. We postulate that higher Hilbert space dimensions tend to overfit the data. Empirically we found that with lower Hilbert space dimensions we have to scale down the dropout appropriately.

As the data is imbalanced between number of positive and negative reviews, we did cost sensitive learning in CoNN-sLDA (Imb) by adjusting the weights of the loss function for different classes and were able to attain slight improvement.

### B.2 Varying number of Iterations of update equations in Algorithm(1)

Figure(6a) shows the plot of varying number of iterations of update equations in algorithm(1) versus the AUC obtained. We can observe the that AUC decreases and the corresponding standard deviation increases as we increase the number of iterations. In our experience, our algorithm works well even for a single iteration and going beyond 5 iteration gives no significant improvement in results.

### B.3 Varying depth of the model

In Algorithm 1, we parameterized the embeddings $\mu_\theta$ and $\mu_{z_i}$ using deep neural networks. Here, we analyze the results of varying the depth of the neural networks and their effect on the corresponding AUC. Figure 6b shows a combination plot, where we visualize the AUC values for various different combinations of depth between $\mu_\theta$ and $\mu_{z_i}$.

We found that two fully connected layers for embedding $\mu_{z_i}$ and a single fully connected layer for embedding $\mu_\theta$ works well for both datasets. Deeper models tend to overfit the data. For training, we recommend starting with a small Hilbert space dimension and batch size, then increase the number of fully connected layers, and finally choose to unroll the model further.

(a) Varying iterations            (b) Varying $\mu_\theta$ and $\mu_{z_i}$'s

Figure 6: (a) Unrolling the model along the x-axis and AUC on the y-axis. For the 'MultiSent' dataset, we found that even using a single iteration works well. (b) Plot showing number of fully connected layers for various combinations of $\mu_\theta$ and $\mu_{z_i}$'s. The AUC values are shown on y-axis. We observe that the setting where there are two fully connected layer of embedding $\mu_{z_i}$ consistently gives good results for varying layers of embedding $\mu_\theta$.

## C. Interpretability: Getting the relevant words based on embeddings obtained

The embedding model defines a relationship between words found in documents $w$ and topic distributions for documents $\mu_\theta$. Usually we calculate the topic distribution from the words in documents. For interpretation purposes, we might wish to go the other direction: from topics $\mu_\theta$ to words in the topic. For instance, after running CoNN-sLDA, we get embeddings for all of the documents. We could then cluster these to get $K$ clusters. We might then ask how to interpret these clusters. We could take the mean embedding of each cluster $\mu_k$ and recover the words that would be associated with the cluster (e.g., SLR, aperture, resolution versus click-and-shoot, special effects). Alternatively, we could run PCA on the embedding space to find the principle directions of variation of document topics. We can then recover the words associated with the end-points of each distribution in order to label this dimension (e.g., light weight versus heavy or easy-to-use versus complicated).

We show here how to define a relationship between a given $\mu_\theta$ and the words associated with the topic. Given the $\mu_\theta$ from CoNN-sLDA model of a document under consideration, we want to find the top $word2vec$ vectors which satisfies the equation(24). If we substitute 14 into 13, we can eliminate the dependence on word topic distributions $z_i$.

$$\mu_\theta = \tanh(\ W_1 \cdot \sum_{i=1}^{N} \{\tanh(\ W_2 \cdot word2vec(w_i) + W_3.\mu_\theta\ )\}\ ) \tag{24}$$

The $\mu_\theta$ terms are related by the sum of the embeddings of words in the text. The embeddings for the same word are always the same, so we can group all embeddings for word class $c$ together and just keep a class weights $F_c$. We set to zero so that we have an equation that measures discrepancy between current system and a consistent system. We then form an objective $J_w$ which is a function of topic distribution $\mu_\theta$ and $K$ word class weights $F_c$.

$$J_w(F_c; \mu_\theta) = \tanh(\ W_1 \cdot \sum_{c=1}^{K} F_c \{\tanh(\ W_2 \cdot word2vec(w_c) + W_3.\mu_\theta\ )\}\ ) - \mu_\theta \tag{25}$$

Minimizing the square of $J_w$ w.r.t. the $F_c$ parameter will give us the weights of the words relevant to the embeddings $\mu_\theta$.

$$F_c^* = argmin_{F_c} J_w^2(F_c; \mu_\theta) \tag{26}$$

We can thus find the top most commonly occurring highly weighted words corresponding to any documents distribution embedding $\mu_\theta$ or examine words associated with any $\mu_\theta$ in the embedded space.