[Reviews · NeurIPS 2018]

Reviewer 1



The authors propose in this manuscript a novel methodology for deep latent representation via joint constraints as an improvement on LDA-derived approaches, w/ applications to text classification. The paper is well written and w/ good scientific content: the idea of transferring the joint constraint structure to a ANN is indeed interesting and theoretically well supported by the mathematical framework. However, the proposed construction requires a number of complex ingredients, and here lies the major flaw of the contribution: the performance gain w.r.t. alternative methods is quite limited (especially in the "20 Newsgroup" dataset, where the difference appears to be statistically not significant). Overall, I would judge the shown experiments as not sufficient to consistently support any superiority claim, rating the method still immature, although promising and of good quality. ### After reading the authors' rebuttal, I changed my rating to a weak accept - the motivations outlined in the rebuttal letter are reasonable and improve the overall shape of the manuscript.

Reviewer 2



Summary: The authors propose a new method that combines a latent Dirichlet allocation (LDA) model with a neural network architecture for the application of supervised text classification –– a model that can be trained end-to-end. In particular, they use a network structure to approximate the intractable inference equations that solve the KL-divergence between the LDA posterior and its approximation which is based on marginal distributions. The authors show that an embedding in a Hilbert space can allow for the approximation of the inference equations, and they choose neural networks to parametrize the functional mapping. Finally, based on two applications, the authors demonstrate an incremental advancement over previous models. Clarity: The overall writing is good, especially as it is a very technical paper with many mathematical details. However, there are some details that cloud the overall impression: - Line 54: the references [4] and [5] do not both refer to the same author. - Equations 2–4: w and z are vectors - Algorithms 1 and 2 suffer from a sloppy typesetting of subscripts, superscripts, and multiplications. - Algorithm 2: o In gradient descent, the gradient should be subtracted and not added. Except r_t is defined as a negative step size ( I could not find the definition for r_t in the paper) o The elements in P^{(0)} have not been defined previously. o The calligraphic T is overloaded, it has been used as an arbitrary operator in Subsection 3.3. Quality: The overall quality of the paper is good. Some additional information could reinforce the reproducibility of the paper, for example more details about how the network is trained. In general, a publication of source code would be very beneficial for the understanding and reproducibility of this work. Originality: To my knowledge, the work is original, and the authors do not merely propose a new architecture based on experimental results but start with theoretical considerations. Significance: The theoretical achievement of approximating the LDA posterior with a PGM is for sure a significant achievement. It would be interesting to know whether the (incremental) performance improvement comes with a tradeoff in terms of computational cost compared to the other models. Minor comments and typos: Line 21: constrain -- >constraint References: inconsistent names (e.g. [13] vs. [16], [20] vs. [21]) and missing journals or online sources ([23], [25]).

Reviewer 3



This paper presents a novel approach for combining neural networks with probabilistic graphical models such as LDA through learning Hilbert space embeddings. The idea is to incorporate domain specific prior structure into the more flexible NN-based learning. There is a growing body of literature in this area, and this work introduces some interesting ideas in this domain. Main strengths: - Well motivated work - Interesting new modeling approach, joint constraint networks - Experimental comparisons with other competing approaches Main weaknesses: - Description of model derivations are vague and grandiose in some places (see below for examples) - Experimental results are not too impressive, and some of the strengths of the proposed model, such as ability to inject prior domain structure and need for reduced training data are not demonstrated effectively The modeling approach is very well motivated with the promise of combining best of both worlds of graphical models and deep learning. Specifically, the need to adapt to limited domain learning problems and to generalize over the vocabulary is certainly important. However, the effectiveness of your model along these dimensions are not effectively demonstrated in the experimental work. For example, it is not clear what domain prior structure is injected into the model, other than selecting the size of Hilbert space dimension. Similarly, the need for reduced training data can be shown by plotting learning curves. I recognize that this is a general missing trend in this area of research, but it would be worth considering some of these additional experiments. Also, given that the results on 20 NG are so close, you might consider also testing your approach on a few other text classification corpora, such as RCV1, and others. Here are some parts of the model description that are vague and require more clarifications or support of evidence: 127-128: although their actual form is complex, maintaining this complete form is unnecessary because it will be different after the embedding anyway. 130: Here, f_1 and f_2 denote the complex dependency structure of equations (5) and (6), respectively 144-146: ..... in order to classify documents accurately naturally causes the embeddings to be injective exactly where they need to be. 169-170: many of the properties and intuitions behind the original LDA model are preserved with the embedded model (this claim needs to be backed by scientific experiments) Typographical/grammatical errors: 21: constraint 114: corresponding probability density